# Experimental and Numerical Simulations of a Solar Air Heater for Maximal Value Addition to Agricultural Products

Zuhair Qamar [1,*], Anjum Munir [2,*], Timothy Langrish [3], Abdul Ghafoor [1] and Muhammad Tahir [4]

1 Department of Farm Machinery & Power, University of Agriculture Faisalabad, Punjab 38000, Pakistan
2 Department of Energy Systems Engineering, University of Agriculture Faisalabad, Punjab 38000, Pakistan
3 School of Chemical and Biomolecular Engineering, The University of Sydney, Camperdown, NSW 2006, Australia
4 Department of Agronomy, University of Agriculture Faisalabad, Punjab 38000, Pakistan
* Correspondence: dr.zqamar@gmail.com (Z.Q.); anjum.munir@uaf.edu.pk (A.M.)

**Abstract:** Agriculture is the backbone of Pakistan's economy. Currently, the agricultural sector is facing many challenges, especially post-harvest losses, which result in lower yield and profitability. These losses may be reduced by developing indigenous post-harvest processing technologies, such as drying out of agricultural products to enhancement of their sustainability and reduce transportation costs. The country has the advantage of an abundant amount of solar insulation, which can be effectively utilized to operate post-harvest machinery, particularly solar heaters and dryers. Currently, conventional solar heaters face challenges due to lower efficiencies. Therefore, in this study, a solar air heater (SAH), having a size $1220 \times 610 \times 65$ mm, was designed and developed to be connected to a milk powder spray drying system for converting raw milk to powder. Computational fluid dynamics (CFD) were used to anticipate air flow and temperature distribution across the SAH to evaluate optimal performance parameters. An air mass flow rate of 0.01 kgs$^{-1}$ was required, with the CFD predicting an outlet temperature of 82 °C compared with the experimental observation of 73 °C. The pressure drop across the SAH was recorded to be 0.0000434 bar at this flow rate, which is negligible, with the SAH operating near ambient pressure. The overall heat loss coefficient for convection was 2.27 W·m$^{-2}$·K$^{-1}$. The energy losses from the SAH were 37% and the useful energy was 63% of the total energy provided to the SAH. The breakeven point of SAH at a minimum of 4 h of daily usage was 3700 h or 2.5 years. The solar air heater used as a preheater for a spray dryer can save 30 PKR·kWh$^{-1}$ of energy.

**Keywords:** solar air heater (SAH); computational fluid dynamics; performance optimization; milk powder; agricultural drying; post-harvest losses





## 1. Introduction

Important concerns that require a linkage strategy include preventing climate change and ensuring a sustainable food supply for an expanding global population. The burden on the agricultural industry and food production has significantly increased because of the growing global population, restricted access to fossil fuels, and higher pricing. Renewable technology in the agricultural sector improves dependability and efficiency by reducing carbon emissions due to the use of fossil-fuel. The technologies adopted in agriculture may include distributed electricity generation, agricultural cultivation greenhouses, space cooling and heating application, the desalination of saltwater, water pumping and irrigation systems, the drying of agricultural products, solar powered agricultural machinery, and farm robots [1].

Pakistan is an agricultural country, with about 67 % of the country's population being directly or indirectly linked to agriculture [2]. The agriculture sector of Pakistan is facing many challenges that are significantly affecting the farming community. The country is

facing a severe energy crisis that is having consequential severe effects on the agriculture sector. Therefore, renewable energy-operated technologies for the processing of agricultural commodities are urgently needed.

Among the largest milk-producing countries, Pakistan ranks as the fifth in the world. Milk production obtained from buffalos and cows was 55 million tons in 2019–2020 [3], compared with the maximum breed potential of 110 million tons. Several factors currently limit the opportunities in the dairy industry, i.e., low returns, restrictions for credit, the low genetic potential of mammals, a lack of proper marketing systems, improper animal care, and a lack of research [4]. A very large amount of milk is wasted due to the absence of better processing facilities. Furthermore, milk powder is imported into the country because of the above-mentioned problems. This problem can be solved by introducing better milk processing facilities.

Milk processing units operated by renewable energy are growing in developing countries that are suffering from an energy crisis. Pakistan is fortunate to have different sources of renewable energy, including solar energy and biomass, so further development of solar thermal energy in Pakistan would be a valuable and viable option. Pakistan is an equatorial country with a very high solar irradiance that ranges from 5 to 7 $kWh \cdot m^{-2} \cdot d^{-1}$ [5], which is available for use in solar heating.

The Efficiencies of the Flat plate collector (FPC) which is applied to transform solar power in thermal energy and the evaluation of the thermal performance greatly depends on how radiation is modelled and assessed. Monte Carlo radiation modelling was used by the authors [6]. This approach makes it possible to analyze spectral and diffuse surfaces as well as complex geometries. Due to the statistical character of this model, a disadvantage is that the surface heat transfer may vary between iterations [7]. The FLUENT program package also contains the surface-to-surface (S2S), discrete ordinates (DO), and discrete transfer radiation models (DTRM). For the investigation of heat transfer inside solar collectors, any one of the three models may be applied. An advantage of the S2S sub model is that it is effective for simulating enclosed radiation heat transfer, while it requires shorter computational times compared with DO and DTRM [8]. Solar radiation, used as an energy source, has been applied as a boundary condition for longwave radiation in the S2S sub model to calculate the heat transfer in previous work [9].

The solar load model provided in ANSYS 2020 R1 FLUENT contains a solar ray tracing method and a surface-to-surface radiation model (S2S). The solar ray tracing technique allows the evaluation of solar energy absorption, while the S2S radiation model assesses the internal distribution of energy. The surfaces exposed to the rays of the sun and consequently absorbing solar thermal energy are the sources. This method of modeling solar load has its limitations. However, the combination of these models should be able to give a realistic representation of an actual case. This S2S model may be used to predict the temperature distribution and the FLUENT model may be used for predicting the airflow patterns.

SAHs are simple, inexpensive, and are widely used in agro-based industries for low-range temperature applications [10]. These solar air heaters absorb radiation and subsequently transmit that energy to fluid flows through collector. SAHs are affordable and are widely used collectors, being used in a variety of applications for solar energy, space heating, wood seasoning, and agricultural drying. Through reviewing previous studies, it was observed that all elements of a SAH, for example, a damping tray, glazing, isolation, extensive surfaces, and the tilting angle, greatly affect the system's thermal efficiency.Deficiencies in the design and production of SAHs may lead to poor performance [11,12]. Jongpluempiti and Pannucharoenwong [13] described the construction of a solar air heater coupled with a spray dryer, and experiments were performed for various inclination angles of the air heater. The results indicated maximum efficiency is given for an angle of 15°, which was equal to the latitude of the site. Aghbashlo and Mobli [14] experimented with powder drying by using a portable solar dryer. The operational cost was negligible compared with that using conventional fuel (wood). The efficiency of the batch dryer was 54% (with solar heating), and it was found that solar drying was cost-effective, since the cost was one-third

of that of wood-fired heating. Fara [15] estimated the performance parameter of an FPC using simulation techniques. Optimal efficiency can be attained by changing the collector area, the storage medium, and the mass flow rate. Tolga and Ural [16] determined that an SAH used in the textile sector has a better performance efficiency than a conventional solar air collector. Nikolic and Lukic [17] investigated the theoretical and experimented performance analysis of FPSC (Flat plate solar collector) which showed a significant increase of 48% in thermal efficiency over a simple solar collector.

Different approaches for enhancing the thermal efficiency of SAHs have been attempted, such as maximizing the size of air-heater modules, using large surfaces of varying shapes and sizes, using sensible or latency storage media, using solar radiation concentrators, and incorporating photovoltaic components in the heaters. In addition, there are several advantages of using SAHs. A conventional solar air heater may use different surface methods in the absorber plate or plates, such as roughened surfaces, extended surfaces, corrugated surfaces, and perforated surfaces to enhance thermal efficiency [18]. As Figure 1 shows, the collector includes an absorber plate and translucent selective cover. The sensible heat capacity of air is less than that of water and thus requires fans with greater flow rates. Black-painted aluminum, GI, and steel are preferred materials for absorber plates, where the plates have high aspect ratios. The cross-sectional area of the SAH duct decreases because of the increase in the aspect ratio. The air velocity and heat transfer rate are also increased by a high aspect ratio [19].

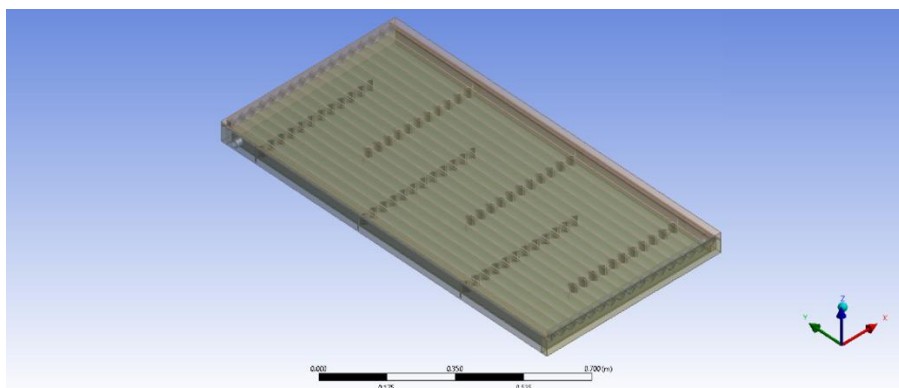

**Figure 1.** Geometric diagram of the solar air heater (SAH).

A transparent cover (glass, polycarbonate sheet) placed above the absorber plate on the top side, and insulation is inserted on the bottom and sides to decrease thermal heat losses. A solar air heater is a type of non-concentrating collector, so it is usually placed in a stationary position with the optimum tilt angle equal to the latitude, north-facing southern hemisphere's and vice versa. In this work, the solar air heater was designed and fabricated at the University of Sydney, Australia with locally and easily available materials.

Gill and Singh [20] created three different types of SAH of single-pass with different covers. The differences affected the heat losses for both convective and radiative heat transfer. The temperature was increased up to 35 °C in the packed bed at an air-flow rate of 0.013 kg s$^{-1}$·m$^{-2}$. The increase in temperature was 12 °C for the first bed with a single glass cover, and it was 18 °C for the second bed with a double cover of glass at an airflow rate of 0.025 kg·s$^{-1}$·m$^{-2}$. A higher efficiency for the SAH was observed with a packed bed (72% at an airflow rate of 0.025 kg·s$^{-1}$·m$^{-2}$) while it was just 45% for the bed with two covers, and 30% for the bed with only one cover.

The aim of this work was to design and develop an SAH to couple with a milk powder spray drying system for converting raw milk to powder. We aimed to use computational fluid dynamics (CFD) to predict the air flow pattern and temperature distribution across the solar air heater to evaluate the best performance parameters. The structure of this paper is as follows. First, we describe the approach to the theoretical modelling technique

used to guide the development of the solar air heater, before describing the experimental equipment (the SAH) that was built and discussing the performance evaluation and further development of the SAH. This is the first time that the performance parameters, i.e., heat transfer rate, air velocity, and baffle arrangements of SAH were optimized using a CFD approach.

## 2. Materials and Methods

The ANSYS 2020 FLUENT software was used to predict the airflow patterns and the S2S model was used for predicting the temperature distribution. A positioning vector was used by the Solar Ray Tracing method to specify the location and amount (power) of the solar energy input, and for characterizing its solar load. A solar calculator was used to state the sun's direction and magnitude. The solar calculator required coordinates or the location of the model, for calculating the vector giving the solar time, the date of the day, and the magnitudes of illumination parameters. The two illumination factors consisted of two irradiation terms, namely diffuse and direct solar irradiation. Transmissivity and absorptivity were used to define three directly visible bands, diffuse hemispherical, and direct IR, while smearing the solar load in cloudy conditions. While using solar ray tracing, the glazing materials were also involved. Reflectivity and transmissivity were specified in the boundary conditions of the wall. The ground reflectivity was set to a value of 0.2.

The collectors for the solar air heaters had a similar design to solar water heater collectors. The solar air heater absorber plate was made of corrugated metal.

It uses a solar-heated heat exchanger with baffles to create a serpentine pathway for the gas. The baffles increase the gas velocity and hence the heat-transfer coefficients from the absorber material to the gas, and the baffles also increase the heat-transfer area between the gas and the absorber surface. The materials were chosen for ease of construction in developing countries, such as Pakistan, so these materials are important due to their availability and low cost, as well as their reasonable heat-transfer performance. The solar air heater was fabricated locally from easily available materials. A schematic diagram of the solar air heaters has been shown in Figure 1. Marine plywood, a medium temperature tolerance wood material, was used in the solar air heater together with Tasmanian oak to make it cheap, portable, and reduce thermal losses. The collector area was 1220 mm × 610 mm with a corrugated iron sheet of 1182 mm × 572 mm as the absorber plate. For maximum absorbance of solar energy, the absorber plate was painted matte black. The optimal air gap was found to be 4–5 cm, so the air gap in the SAH was assumed to be 4 cm, because the polycarbonate sheet was flexible. Plywood having dimensions of 1220 mm × 610 mm × 6 mm has been used as a backing sheet, and fiberglass insulation of 30 mm thickness was used between the corrugated iron and the plywood, to reduce heat losses. Transparent cellular polycarbonate sheets of dimensions 1200 mm × 610 mm × 8 mm were used instead of brittle glass. These polycarbonate sheets provided good thermal insulation and have high strength, as well as being lighter in weight than glass. The airflow passage between the absorber plate and the glazing (polycarbonate sheet) was 30 mm. The cellular polycarbonate sheet had a light transmission of about 80%. The solar heat gain coefficient was 0.80, and the heat transfer coefficient for heat loss ($U$) was 2.9 $\mathrm{Wm^{-2}\,K^{-1}}$. The polycarbonate sheet retains good light transmission properties for over ten years. The polycarbonate sheets were more durable and tolerated higher temperatures than polythene plastic film glass having a higher transmissivity (90%). Polythene sheets have been frequently used in SAHs, but they become more brittle over time than polycarbonate. Bakari and Minja [21] explained the characteristics of the top covers for the solar air heaters. A good top cover should have low reflection, absorption, and high transmission of light. Glass has 90% solar irradiance transmitting ability while plastic has some advantages over glass such as being less likely to break, having a lighter weight, and being cheaper. Plastic cannot withstand the high temperatures when the collector is idle, and UV light affects the transmissivity as well. Polystyrene foam insulation was used at the sides and the back of the SAH to reduce heat losses.

Kaushal and Goel [22] explained that the conventional SAHs (flat plate) have poor thermal performances as heat transfer coefficients between the absorber plates and the flowing medium (air) are typically low. The heat-transfer coefficient can be enhanced with artificial roughness which can create turbulent flow which has a positive impact on heat transfer. Based on a literature review, it was found in different research that the Reynolds numbers for solar air heaters range from 2000 to 16,000 [23,24].

### 2.1. Thermal Analysis of the Solar Air Heater

The efficiency of the solar air heater (SAH) has been assessed using Equation (1). The following parameters were calculated using Equations (2)–(10): the energy gain ($Q_u$), the mass flow rate, the equivalent diameter, the Reynolds number, the tilt angle, the friction factor for laminar or turbulent flow, the pressure drops, and the overall heat transfer coefficient. The thermal performance evaluation parameters of the SAH are presented in Table 1.

**Table 1.** Thermal properties of the solar air heater, [25].

| Parameter | Equation | Equation No. |
|---|---|---|
| Efficiency of SAH | $\eta = \dfrac{Q_u}{I_r\,A_c}$ | (1) |
| Energy Gain | $Q_u = \dot{m}C_P(T_{out} - T_{in})$ | (2) |
| Mass flow rate | $\dot{m} = \rho.A_C.V$ | (3) |
| Equivalent Diameter | $de = \dfrac{2LH}{L+H}$ | (4) |
| Reynolds Number | $Re = \dfrac{V.\rho.2LH}{\mu(L+H)}$ | (5) |
| Tilt angle | $\beta = L \pm 15^\circ$ | (6) |
| Friction factor | $f = \dfrac{24}{Re}$ (for laminar) | (7) |
| | $f = 0.079\,Re^{-0.25}$ (for turbulent) | (8) |
| Pressure Drop | $\Delta P = \dfrac{2f\left(\rho.V^2\right)L}{de}$ | (9) |
| Overall heat transfer coefficient | $h = \dfrac{Q_u}{A_C\left(T_P\_T_o\right)}$ | (10) |

### 2.2. Heat Losses

The total energy flow rate in the SAH is the sum of the useful energy flow rate $Q_u$, the stored energy flow rate $Q_s$, which is neglected because no storage material is in use and the system is considered to be close to steady state, $Q_{loss,th}$ is the thermal loss rate, and $Q_{loss,opt}$ is the optical loss and $U$ is the overall heat loss coefficient, which can be determined using Equations (11) to (16), respectively. The optical losses are caused by the cover transmission losses and the absorption ability of the absorber, the so-called transmission absorption product $(\tau\alpha)_e$. The optical losses are calculated when the SAH is exposed to solar light. The heat temperature $T_{heater}$ is assumed here to be the average value of the inlet and outlet temperatures. The heat loss parameters of the solar air heater are presented in Table 2.

**Table 2.** Heat loss parameters of the solar air heater, [26].

| Parameter | Equation | Equation No. |
|---|---|---|
| Total energy flow | $G = Q_u + Q_s + Q_{loss,th} + Q_{loss,\,opt}$ | (11) |
| Heat loss | $Q_{loss} = C_p.m.(T_{in} - T_{out})$ | (12) |
| Overall heat loss coefficient (W/m$^2$ K) | $U_L = \dfrac{Q_{loss}}{A_C\left(T_P-T_a\right)}$ | (13) |
| Thermal heat loss | $Q_{loss,th} = U_L A_c(T_{heater} - T_a)$ | (14) |
| Optical heat loss | $Q_{loss,opt} = G(1 - (\tau\alpha)_e)$ | (15) |
| Efficient transmission and absorption of the product | $(\tau\alpha)_e = \dfrac{Q_u + U_L A_C\left(T_P - T_a\right)}{I_t.A_c}$ | (16) |

*2.3. Economic Evaluation*

Management of machinery includes two important elements of finance, which are the ownership or capital cost (fixed) and the operating cost, which occurs during the use of the machinery [27]. Different factors affect the fixed cost, including depreciation, interest, and taxes. Various methods may be used to determine depreciation, including the straight-line method, which is one of the most straightforward ones and has been used here [28]. Salvage value, which is 10–15% of the capital cost, has been assigned to the machine [27]. Here, the salvage value from the first installed cost, and difference was divided by useful life (years) for the machinery [29].

$$D_s = \frac{(P - S)}{L} \tag{17}$$

where $D_s$ = depreciation amount per year by the straight-line method in rupees, $P$ = purchase value (capital cost) in rupees, $S$ = salvage value in rupees, and $L$ = useful life in years.

The cost of machinery also includes the interest cost. Interest is also considered an opportunity, and the interest on the machinery may be calculated by the equation developed by Jacobs and William [27].

$$I = \frac{(P + S)}{2 \times i} \tag{18}$$

where

I = annual interest amount in rupees, and

$i$ = interest rate as a fraction, and $P$ and $S$ have been defined above.

## 3. Results

The structure of this section is to report and discuss the theoretical performance of the SAH in Section 3.1, before discussing the actual heat losses and leakages from the experimental unit in Section 3.2. The complete technical performance of the SAH and its further development follow in Section 3.3, building on the important preliminary theoretical work in Section 3.1 and practical foundations in Section 3.2. To complete a techno-economic assessment of the SAH, the economics of the SAH are presented and discussed in Section 3.4.

*3.1. Solar Air Heater: Theoretical Performance*

This section discusses the results of the computational and theoretical performance of a solar heater containing the corrugated absorber plate with a baffled SAH, with respect to varying mass flow rates, solar intensities, and geometries. This section indicates the optimal results obtained at appropriate mass flow rates for a solar-assisted spray-drying process. The airflow analysis and temperature distribution of the SAH was conducted using the FLUENT software [30,31]. The key advantage of FLUENT is predicting the detailed air flow patterns in the whole equipment. Baffles were used because they gave higher air velocities and higher heat-transfer coefficients compared with arrangements without baffles in the solar air heater. The entire structure above absorber plate with baffles was meshed in triangular elements and rectangular meshing was used on the baffles. Triangular meshing was chosen rather than a hexagonal mesh because it is finer and gives a better prediction of results in the complicated regions shown in Figure 2. In computational analysis, the solar irradiance applied on the absorber plate of the SAH was 900 W·m$^{-2}$. The velocity flow rates for the air are also shown in Figure 3, which indicates that the velocity of the inlet and outlet air flows was the same and very little pressure drop was recorded in the equipment. Here, the color variation in the predicted temperature distributions for the solar air heater showed that the air temperature increased when the air moved towards the outlet in Figure 4. The temperature distribution in the solar air heater, its maximum plate temperature was computed to be 102 °C based on the boundary conditions, and the outlet temperature was predicted to be 82 °C for a mass flow rate of 0.01 kg·s$^{-1}$. The baffles were equally distributed over the plate which kept the air velocity high, giving more heat transfer than without the baffles. After analyzing the situation at different

flow rates (0.005 kg·s$^{-1}$, 0.01 kg·s$^{-1}$), 0.01 kg·s$^{-1}$, a mass flow rate was chosen, which was also suitable for the spray drying system. A greater radiation intensity increases the outlet temperature which helps to increase the collector efficiency. The variation in the collector efficiency at different mass flow rates was connected with the changing outlet temperatures and geometries. The computational method predicts the outlet temperature which can be used to assess the efficiency of the system [32,33]. The theoretical efficiency of the solar air heater was calculated from the experimental data to be 69%; while using the computational method, the solar air heater efficiency was predicted to be 73%. To obtain the computational results, the input parameters were the inlet temperature (35 °C) and solar irradiance (950 W·m$^{-2}$). The experimental value for the outlet temperature was 82 °C and the actual thermal energy gain was calculated to be 473 W. These experimental results are close to the computational predictions, and the experiments included a 38 °C inlet temperature and 327 W of thermal gain. The difference in the rate of thermal gain is due to the different solar irradiance values of 700 W·m$^{-2}$ and 950 W·m$^{-2}$ for the actual and the computational experiments, respectively. The details of the actual laboratory experiment are summarized later in this paper.

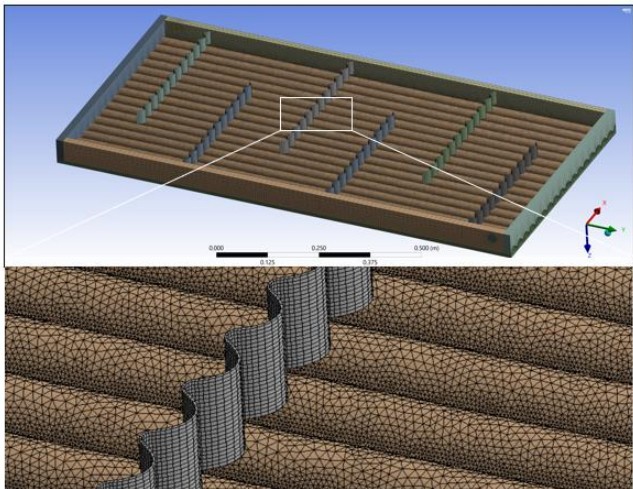

**Figure 2.** Surface mesh for the solar air heater.

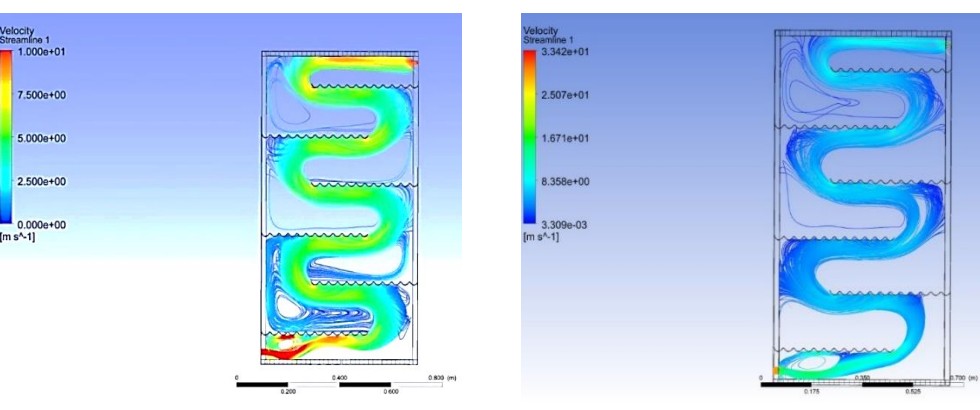

**At mass flow rate 0.01 kg·s$^{-1}$**      **At mass flow rate 0.005 kg·s$^{-1}$**

**Figure 3.** Performance prediction of an air-flow distribution pattern using CFD for the SAH (e$^x$ = 10$^x$).

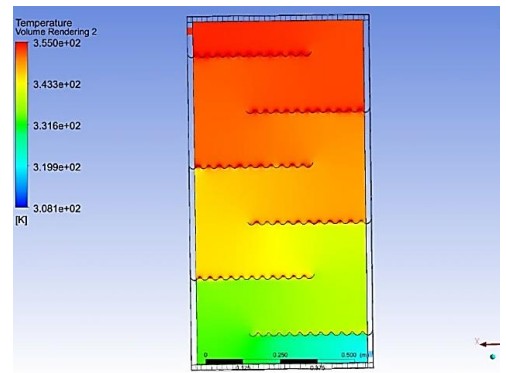

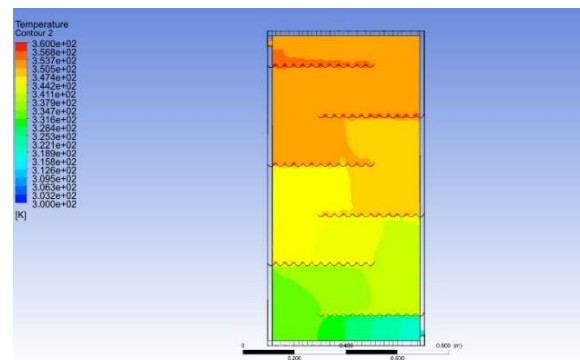

At mass flow rate 0.01 kg·s⁻¹                    At mass flow rate 0.005 kg·s⁻¹

**Figure 4.** Performance prediction of air temperature distribution pattern using CFD for the SAH ($e^x = 10^x$).

The CFD simulation with the baffles predicts a maximum collector efficiency of about 73%, and the temperature increase in the solar air heater with baffles results in better performance, due to the increased rate of heat transfer from the absorber plate to the flowing medium (air) [26,34]. The computational method predicts that this design of solar air heater has a slightly higher efficiency than the actual observation.

The fine meshing resulted in almost 84 K cells for the finest mesh. The pattern of the baffles on the corrugated SAH increases the heat transfer area as well as the heat transfer rate.

*3.2. Heat Losses from the Solar Air Heater*

Having assessed the theoretical performance of the solar air heater using CFD, the experimental aspects of its evaluation are now reported and discussed. The first aspect reported in this section, is the area of heat losses. An experiment was performed over one hour to achieve steady-state conditions, as shown in Figure 5 for the experiments in the laboratory (with no solar energy input) and Figure 6 for the experiments in the courtyard (with solar energy input). The ambient, inlet, outlet, and plate temperatures measured in the laboratory with no energy input were 26 °C, 59 °C, 48 °C, and 57 °C, respectively. From Equations (11)–(16), the overall heat loss coefficient for convection was then estimated as 4.9 W·m⁻²·K⁻¹ using Equation (13). The radiation and convective losses in the SAH were calculated in the courtyard where solar energy was used as the input energy. The thermal losses were then estimated as 154 W using Equation (14), and the optical losses were estimated as 46 W using Equation (15). The total losses were estimated to be 200 W using Equation (11). The useful energy flow rate provided to the air when the experiment was performed in the courtyard was 334 W using Equation (2). The losses were 37% and useful energy was 63% of the total energy provided by the SAH. The effective transmittance absorbance product $(\tau\alpha)_e$ of the SAH was 91% (Equation (16)) for this system with a double polycarbonate sheet at the top cover and a corrugated iron sheet with baffles at a spacing between the baffles of 17 cm.

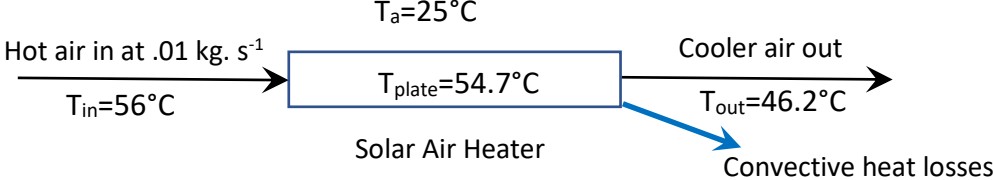

**Figure 5.** Schematic diagram of mass and energy flows in the experiments for the heat loss measurements in the laboratory.

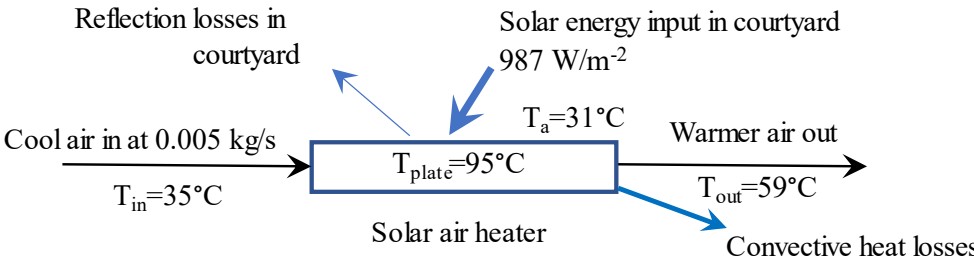

**Figure 6.** Schematic diagram of mass and energy flows in the experiments for the heat loss measurements in the courtyard.

In Figure 5, the temperature measurements in the laboratory (with no solar energy input) are given in Table 3 for a one-hour experiment to achieve steady-state conditions and a heat input from an electric heater to the air entering the SAH. The ambient, inlet, outlet, and plate temperatures measured in the laboratory with no energy input were 26 °C, 58 °C, 47 °C, and 54 °C, respectively. The overall heat loss coefficient for convection was then estimated as 4.9 $W \cdot m^{-2} \cdot K^{-1}$. In this experiment, no polystyrene and double poly-carbonate cover sheet was used. The spray dryer was attached for air circulation with a mass flow rate of 0.01 $kg \cdot s^{-1}$. The aspirator gave an air flow of 28.5 $m^3 \cdot h^{-1}$. In addition, the laboratory test checked for any significant air leakage from the solar air heater, and no significant sources of leakage were found. The above two experiments show the results from a solar air heater in which the convective heat losses were calculated. The experiment was performed in a laboratory of the School of Chemical and Biomolecular Engineering, The University of Sydney, Australia.

**Table 3.** Temperature measurements in the laboratory for heat loss calculations.

| | Inlet Temperature (°C) | Outlet Temperature (°C) | Ambient Temperature (°C) | Plate Temperature (°C) | Mass Flow Rate (kg/s) |
|---|---|---|---|---|---|
| Experiment 1 | 58 | 47 | 26 | 54 | 0.01 |
| Experiment 2 | 56 | 46.2 | 25 | 54.7 | 0.01 |

Having assessed the heat losses, the overall performance evaluation and the further development of the design will be reported and discussed in the next section, Section 3.3.

### 3.3. Performance Evaluation and SAH Devlopment

The performance evaluation of the solar air heaters without baffles was investigated. The experiment was performed in the courtyard using a blower, having an air flow rate of 0.005 $kg \cdot s^{-1}$, in an open loop mode as shown in Figure 6. The experiment was performed during peak sunlight hours, and the average solar irradiance was recorded to be 987 $W \cdot m^{-2}$. The very large distinction between outlet air temperature and the plate temperature is shown in Figure 6.

The thermal losses of the solar air heater were high, since there was only one polycarbonate sheet, and there were no polystyrene sheets on the back and sides in this experiment. Due to excessive heat losses, the thermal efficiency of the system was low, and low amounts of useful energy were produced.

Figure 7 shows a graphical representation of the temperature of the air inlet, the air outlet, and the plate temperature over a time period of one hour. The solar irradiance is shown on the right axis and has a gradually declining trend. However, the use of baffles increases the air velocity inside the heating chamber and enhances the heat transfer rate. In addition, significantly reduced thermal losses were found when good insulation was provided. The analysis showed that the plate temperature attained a steady state condition after 25 min, as shown in Figure 7.

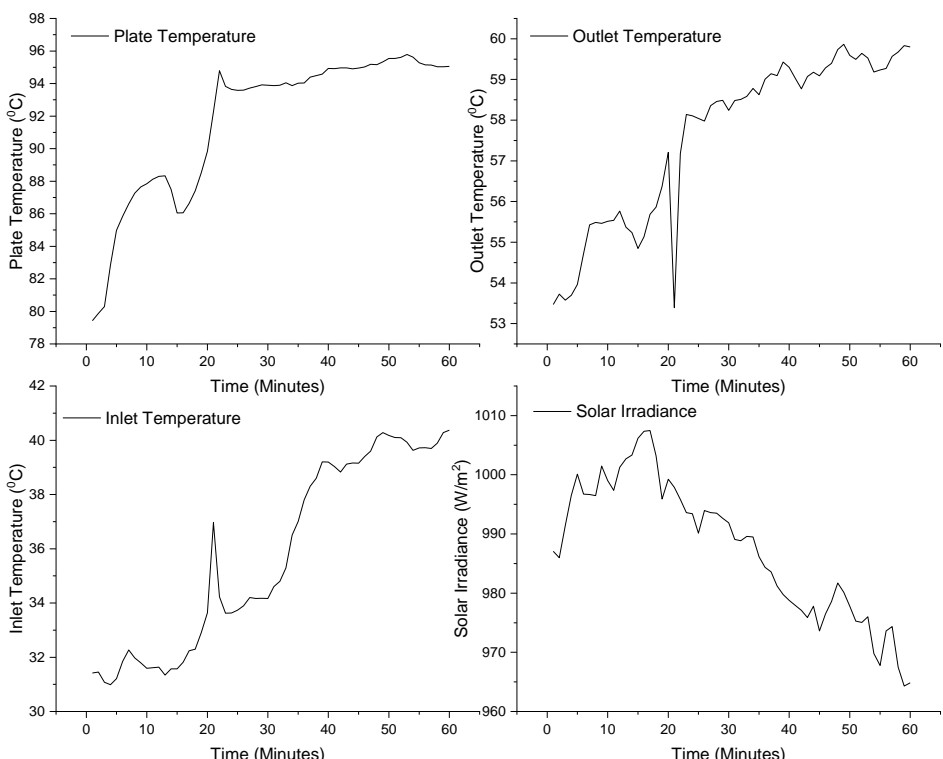

**Figure 7.** Graphical representation of different points temperatures and solar irradiance for parametric study of the solar air heater.

A performance evaluation of the solar air heaters with baffles and insulation was also investigated and is presented in Figure 8.

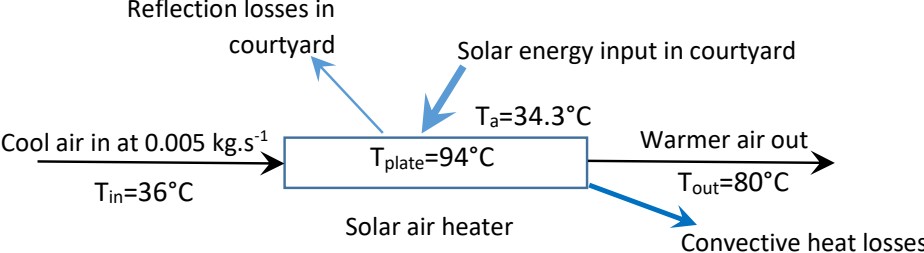

**Figure 8.** Schematic diagram of mass and energy flows in the experiments for performance evaluation and the heat loss measurements in the courtyard.

The experiment was performed in the courtyard using a blower, having an air flow rate of 0.005 kg·s$^{-1}$ in open loop mode, as shown in Figures 8 and 9. The experimental setup was installed twice in the courtyard to get good results after insulating the sides and back of the solar air heater with polystyrene. The solar irradiance was measured to be, on average, at 740 W·m$^{-2}$ and 840 W·m$^{-2}$ on two days of experiments. The difference between the outlet air temperature and the plate temperature was reduced, compared with the previous experiment. Figure 8 represents the different point temperatures and solar irradiance for the experiments over a period of one hour at steady state. A comparison of the results has shown that there was an increase in the outlet air temperature and a reduction in the difference between plate temperature and the outlet air temperature. These results indicate that the heat transfer rate for a corrugated, black-painted, iron sheet with the working fluid (air) increased due to greater air velocities with the provision of baffle plates. In this experiment, the thermal heat losses from the solar air heaters were reduced from 4.9 to 2.27 W·m$^{-2}$·K$^{-1}$, because the back and the sides of the SAH were provided with appropriate insulation material (polystyrene sheet; thermal conductivity 0.033 W·m$^{-1}$·K$^{-1}$).

Due to the reduction in the heat losses, the thermal efficiency of the system was increased, and the system produced more useful energy. The mass flow rate in this experiment was low, which had a direct effect on the solar air heater efficiency. The use of obstacles or baffles enhanced the heat transfer rate, and the heat losses were significantly reduced with good insulation. The analysis showed, in both experiments, that the experimentally measured temperatures were effectively at steady state conditions during the whole trial.

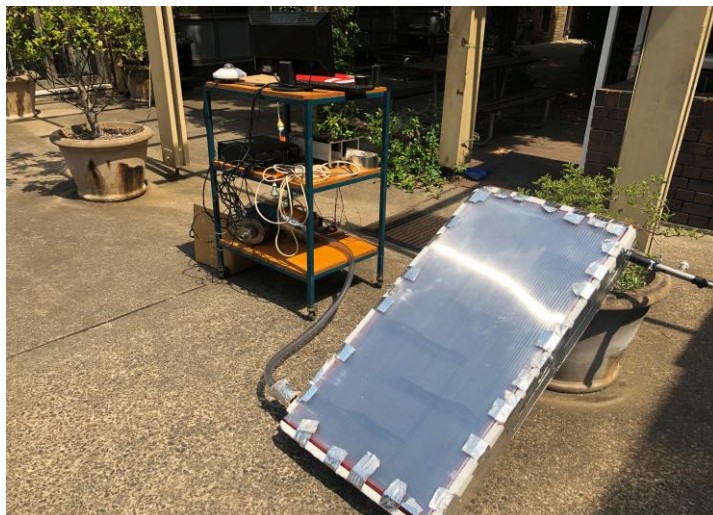

**Figure 9.** Experimental setup for evaluating performance parameters of SAH.

In Figure 10, the solar irradiance value dropped to 250 W·m$^{-2}$ due to some clouds moving across the SAH, and the outlet and plate temperature also showed some variation at that specific time. The radiative and convective losses in the SAH were calculated in the courtyard where solar energy was used as input energy. The thermal losses were then estimated as 154 W, and the optical losses were estimated as 46 W. The total losses were estimated to be 200 W. When the experiments were performed, the useful energy flow rate provided to the SAH outlet air in the courtyard experiment was 334 W. The losses were calculated to be 37%, and the useful energy was estimated to be 63% of the total energy provided to the SAH. The effective transmittance absorbance product $(\tau\alpha)_e$ of the SAH was calculated to be 81%. The system had a double polycarbonate sheet at the top cover and a corrugated iron sheet with baffles at a spacing between baffles of 17 cm. According to the findings of El-Said [35], the overall thermal performance of his air heater was approximately 77% at an air mass flow rate of 0.03 kg·s$^{-1}$, an opening diameter of 3 mm, and a baffle angle of 7°. The CFD-predicted pressure drops across this current solar air heater varied between 0.0001505 and 0.000017 bar. These results are significant for the application of this enhanced solar air heater and demonstrate that the corrugated perforated baffles in the air channel are an effective design feature that improves the collector efficiency. In the above section, the results for the actual pressure drop were very similar, recorded to be 0.0000434 bar for a system operating near ambient pressure (1.013 bar), so the pressure drop was negligible.

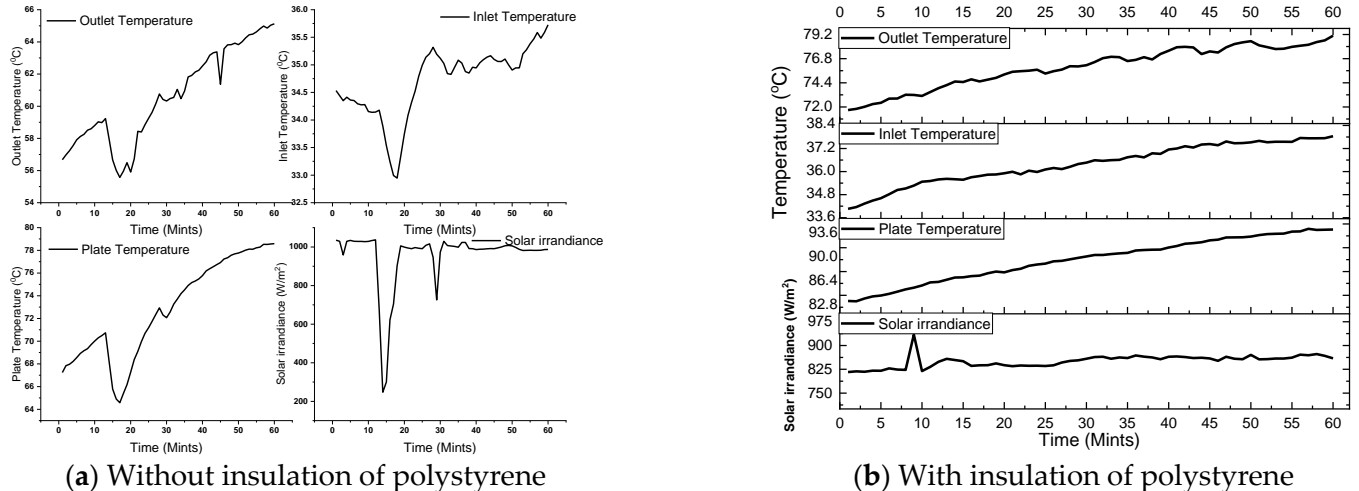

(**a**) Without insulation of polystyrene　　　　　(**b**) With insulation of polystyrene

**Figure 10.** Graphical representation of different points temperatures and solar irradiance for parametric study of the solar air heater at mass flow rate of 0.005 kg·s$^{-1}$.

The final experiment was conducted in the courtyard of the School of Chemical and Biomolecular Engineering the University of Sydney, Australia (33.8905° S, 151.1929° E). The solar air heater was used as a pre-heater providing an outlet air temperature of 73 °C with a humid enthalpy of 164 kJ·kg$^{-1}$. The mass flow rate was 0.01 kg·s$^{-1}$

The ambient temperature measured was 31 °C. The energy flow rate into the electrical heater was 0.327 kW in the experiment using the solar air heater. Figure 11 shows the inlet temperature, the outlet temperature, the plate temperature, and the solar irradiance for the final experiment using a the solar-assisted spray drying system. The inlet temperature shows a steady value, while the plate and outlet temperatures dropped in the initial 5 min then reached relatively steady state conditions. The average outlet temperature under steady state conditions was measured to be 73 °C. A greater difference between the inlet and outlet air temperatures indicates the good efficiency of the solar air heater. The lower difference between the plate temperature and the outlet temperature indicated a good heat transfer rate. On the right *y*-axis, the solar irradiance was seen to rise, and fall. According to international standards, for a solar based experiment to be considered valid, the solar irradiance value cannot be increased or decreased by more than 100 W·m$^{-2}$ over a ten minute time interval, and a solar irradiance value lower than 450 W·m$^{-2}$ or greater than 1100 W·m$^{-2}$ also render the experiment invalid. In addition, the experiment for performance evaluation must be conducted between 10.00 to 14.00 solar time for the experiment to be considered valid [36]. These validity requirements were all met in these experiments. The Graphical representation of the below experiment for one hour is shown in Figure 12.

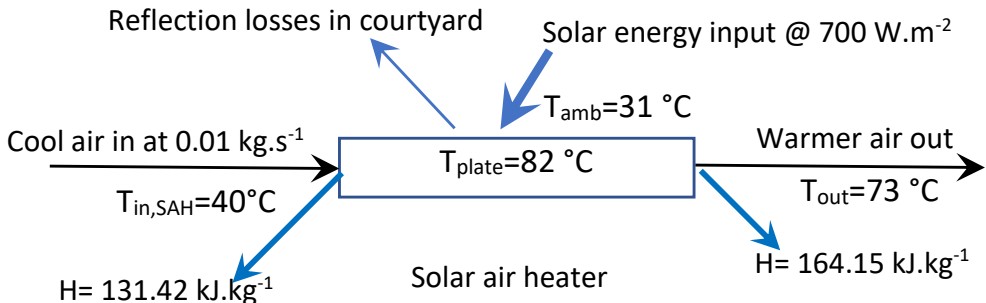

**Figure 11.** Schematic diagram of mass and energy flows in the experiments for performance evaluation and the heat loss measurements in the courtyard at mass flow rate of 0.01 kg·s$^{-1}$.

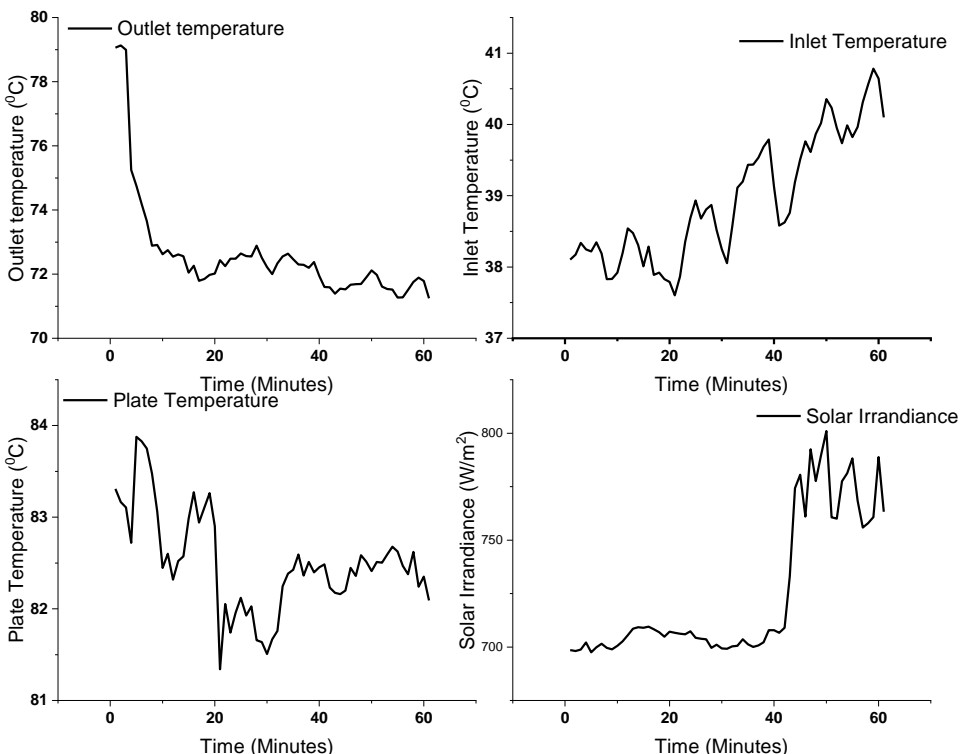

**Figure 12.** Graphical representation of the different temperatures and solar irradiance as function of time during the final experiment.

### 3.4. Economic Analysis

With the technical performance of the SAH being confirmed to be satisfactory, the economic assessment will now be given, to complete a techno-economic assessment of the SAH. The following assumptions were made during the economic cost analysis of the solar-assisted spray drying unit.

The life span of the solar-assisted spray dryer = 15 years

Daily usage = 4 h

Total utilized hours over the equipment life span = 21,900 h

The initial cost of complete solar air heater = P = 13,000 PKR

Salvage value, *S* (10% of the initial cost of solar air heater) = 1300 PKR

Fixed cost: The straight-line method below was used for calculating depreciation cost ($D_s$).

$D_s = \frac{13,000-1300}{15}$ = 780 PKR per year = 0.09 PKR per hour

Interest cost = $0.08 \times \frac{13,000+1300}{2}$ = 572 PKR per year = 0.065 PKR per hour

Total fixed cost = $D_s + I + P$ = 0.09 + 0.065 + 13,000 = 0.155 PKR per hour

Variable cost:

Repair and maintenance = 2.5% of SAH = $0.025 \times 13,000$ PKR = 325 PKR/year = 0.23 PKR/h

Electricity cost per unit = 30 PKR·kWh$^{-1}$

Total cost for forced circulation of air = 0.2 kW × 30 PKR·kWh$^{-1}$ = 6 PKR·h$^{-1}$

The total variable cost of the spray drying system = 0.23 + 6 = 6.23 PKR·h$^{-1}$

The total cost of SAH = Total Fixed cost + Total variable cost = 0.155 + 6.23 = 6.38 PKR·h$^{-1}$

### 3.5. Breakeven Analysis

The economic viability of any organization often starts with a breakeven analysis. In this analysis, the focus is on finding the operation level where the organization has no profit at all. This analysis is a very important step when launching a new product for any

industry. Through this analysis, a reference may be created for further operations as this helps in indicating those levels of operation where cost and revenue become equal. If a breakeven analysis is conducted for this design of solar air heater, which works for about 4 h a day, then in 3700 h, or in other words in 2.5 years, the breakeven point is achievable. About 30 PKR/kWh of energy is saved by this solar air heater, which works as a preheater for the spray dryer. The breakeven analysis of the solar air heater is shown in Figure 13.

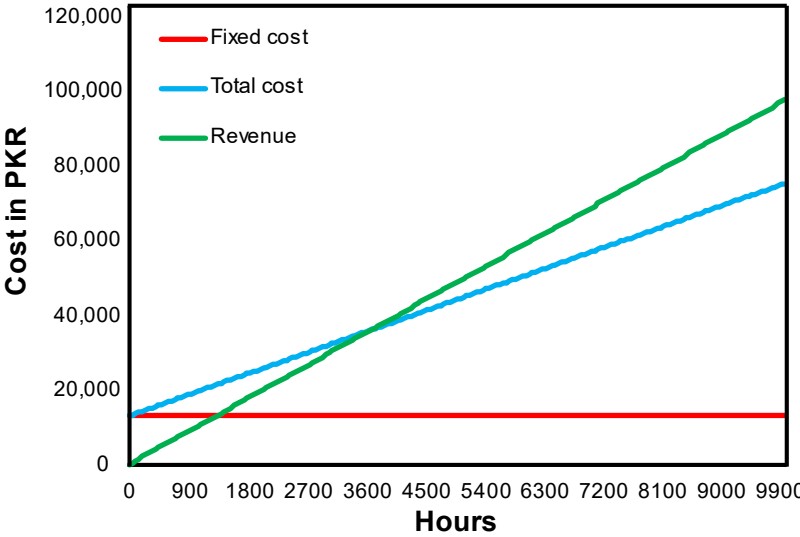

**Figure 13.** Breakeven analysis of the solar air heater.

## 4. Discussion

The radiative and convective losses in the SAH were calculated in the courtyard where solar energy was used as input energy. The thermal losses were then estimated as 154 W, and the optical losses were estimated as 46 W. The total losses were estimated to be 200 W. When the experiments were performed, the useful energy flow rate provided to the air in the courtyard was 334 W. The losses were calculated to be 37% and the useful energy was determined to be 63% of the total energy provided to the SAH. The effective transmittance absorbance product $(\tau\alpha)_e$ of the SAH was calculated to be 81%. The system included a double polycarbonate sheet as the top cover and a corrugated iron sheet with baffles at a spacing between baffles of 17 cm. This thermal performance may be compared with the SAH of El-Said [35] where this device had an efficiency of approximately 77% at an air mass flow rate of 0.03 kg·s$^{-1}$, an opening diameter of 3 mm, and a baffle angle of 7°. The experimental pressure drops across this SAH varied between 0.0001505 and 0.000017 bar.

In the above paper, the results showed a thermal efficiency of 63% for the single-pass roughened-surface SAH while the thermal efficiency was 77% for the double-pass SAH. The pressure drop across the SAH here was found to be 0.0000434 bar, and the agreement between the CFD predicted pressure drop and that measured experimentally was satisfactory. The solar-assisted spray dryer operated at atmospheric pressure (except for the compressed air to the atomizer at 5 bar), so the SAH pressure drop is negligible. These CFD numerical results are significant for the application of this enhanced SAH and demonstrate that the corrugated, perforated baffles in the air channel are an effective factor in improving collector efficiency.

## 5. Conclusions

This study is focused on design optimization and parametric studies of a solar air heater which has been conducted through an experimental study and a numerical prediction of the SAH performance. The Following conclusions can be extracted from this work.

There is very good agreement between the experimental results and the CFD predictions for the increase in the SAH outlet temperature. Despite some minor differences, the

CFD simulation tool has predicted a similar SAH performance to that which was observed, so the CFD approach may be applied in the future for more solar air heater applications. The effect of variable solar irradiance and mass flow rate values on the thermal efficiency of a solar air heater has also been investigated. The result shows a direct relationship between the solar irradiance, the mass flow rate, and the thermal efficiency of the system. The experimental study showed that 0.327 kW was produced and thermal heat losses from the solar air heater were reduced from 4.9 to 2.27 $W \cdot m^{-2} \cdot K^{-1}$ because the back and sides of SAH were appropriately insulated (polystyrene sheet; thermal conductivity 0.033 $W \cdot m^{-1} \cdot K^{-1}$). The pressure drop across the SAH was recorded to be 0.0000434 bar; therefore, the pressure drop is negligible.

**Author Contributions:** Conceptualization, Z.Q. and A.M.; Methodology, Z.Q. and T.L.; Software, Z.Q.; Validation, Z.Q. and A.M.; Writing—original draft, Z.Q. and A.M.; Writing—review and editing, Z.Q., T.L., A.G., and M.T.; Project administration, A.M. and T.L.; Funding acquisition, Z.Q. and A.M. All authors have read and agreed to the published version of the manuscript.

**Funding:** This research was funded by the Higher Education Commission of Pakistan.

**Institutional Review Board Statement:** This study is a mandatory part of my PhD studies and all other authors are the member of my PhD supervisory com-mittee. Hence there is no conflict of interest and lack of ethics in publishing this manuscript.

**Data Availability Statement:** The research profile and the data of my supervisory committee (Co authors) have been given in the links below. http://www.uaf.edu.pk/EmployeeDetail.aspx?userid=711; https://www.sydney.edu.au/engineering/about/our-people/academic-staff/timothy-langrish.html; http://www.uaf.edu.pk/EmployeeDetail.aspx?userid=387; http://uaf.edu.pk/EmployeeDetail.aspx?userid=158; For more research details, google scholar and researchgate sites of the individuals can be seen.

**Acknowledgments:** The authors acknowledge Financial and Technical support from the Higher Education Commission of Pakistan (HEC), University of Agriculture, Faisalabad Pakistan, and the School of Chemical and Biomolecular Engineering for this research, The University of Sydney is also acknowledged for support and guidance for this research.

**Conflicts of Interest:** The authors declare no conflict of interest.

### Nomenclature

$\dot{m}$ = mass flow rate of air ($kg \cdot s^{-1}$)
$C_p$ = specific heat capacity of air ($kJ \cdot kg^{-1} \cdot K^{-1}$))
$T_{out}$ = outlet air temperature (°C)
$T_{in}$ = inlet air temperature (°C)
$I_r$ = solar irradiance ($W \cdot m^{-2}$)
$A_c$ = area of the collector ($m^2$)
$\rho$ = density of the fluid ($kg \cdot m^{-3}$)
$V$ = velocity of air ($m \cdot s^{-1}$)
$L$ = length of SAH (m)
$H$ = height of the SAH (m)
$V$ = velocity of the air in SAH ($m \cdot s^{-1}$)
$\mu$ = dynamic viscosity ($kg \cdot s^{-1} \cdot m^{-1}$)
$\beta$ = collectors tilt angle (degrees)
$L$ = latitude of the specific site (degrees)
$Q_u$ = useful energy flow rate (W),
$Q_s$ = stored energy flow rate (W),
$Q_{loss,th}$ = thermal loss rate (W),
$Q_{loss,opt}$ = optical loss (W),
$(\tau\alpha)_e$ = transmission absorption product
FPSC = Flat plate solar collector
SAH = Solar air heater

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
