# Peer review of "Experimental and Numerical Simulations of a Solar Air Heater for Maximal Value Addition to Agricultural Products"

_agriculture, doi:10.3390/agriculture13020387_

Round 1
Reviewer 1 Report
Introduction: Please address the recent advances in renewable energy-operated technologies in agricultural field across the world, and then introduce Pakistan.
Throughout the introductory section and the later part, little discussion has been made pertaining to various S2S models. Please cite more appropriate references.
Author Response
Q.1 Introduction: Please address the recent advances in renewable energy-operated technologies in the agricultural field across the world, and then introduce Pakistan.
Ans.
A paragraph (with reference) on “Recent advances in renewable energy-operated technologies in the agricultural field across the world” has been added in the Introduction Chapter. The paragraph is also produced here.
“Important concerns that require a linkage strategy are preventing climate change and ensuring a sustainable food supply for an expanding global population. The burden on the agricultural industry and food production has significantly increased because of the growing global population, restricted access to fossil fuels, and higher pricing. Renewable technology in the agricultural sector improves dependability and efficiency by reducing carbon emissions from fossil fuels. The technologies adopted in agriculture were distributed electricity generation, agricultural cultivation greenhouses, space cooling, and heating application, desalination of saltwater, water pumping and irrigation systems, drying of agricultural products, solar powered agricultural machinery, and farm robots.”
- 2 Throughout the introductory section and the later part, little discussion has been made pertaining to various S2S models. Please cite more appropriate references.
Ans. 2 The more appropriate references related to CFD sub-models are added on line 71 to line 83 and are produced as under:
“The efficiency of the FPC and evaluation of the thermal performance greatly depends on how radiation is modelled and assessed. Monte Carlo radiation modelling was used by the authors [1]. This approach makes it possible to analyze spectral and diffuse surfaces as well as complex geometries. Due to the statistical character of this model, a disadvantage is that the surface heat transfer may be varied between iterations [2]. The FLUENT program package also contains the surface-to-surface (S2S), discrete ordinates (DO), and discrete transfer radiation models (DTRM). For the investigation of heat transfer inside solar collectors, any one of the three models might be applied. The advantages of S2S sub-model is effective for simulating enclosed radiation heat transfer without involving any mediums and it required shorter computational time as compared to DO and DTRM [3]. The authors investigated heat transfer across solar air collectors. The Solar radiation used as an energy source applied as a boundary condition for longwave radiations in S2S sub-model to calculate the heat transfer [4].”
- Sultana, T., et al., Numerical and experimental study of a solar micro concentrating collector. Solar Energy, 2015. 112: p. 20-29.
- Howell, J.R., et al., Thermal radiation heat transfer. 2020: CRC press.
- Filipović, P., et al., Numerical and experimental approach for evaluation of thermal performances of a polymer solar collector. Renewable and Sustainable Energy Reviews, 2019. 112: p. 127-139.
- Reichl, C., et al., Comparison of modelled heat transfer and fluid dynamics of a flat plate solar air heating collector towards experimental data. Solar Energy, 2015. 120: p. 450-463.

Reviewer 2 Report
See attachment

Author Response
Q.1 Problem is well defined in the abstract, but I suggest showing the novelty of the design.
Ans. 1: The novelty of the design has been included in the revised version of the article at the end of the introduction. “The aim of this work was to design and develop a drying SAH to couple with a milk powder spray drying system for converting raw milk to powder. We aimed to use computational fluid dynamics (CFD) to predict the airflow pattern and temperature distribution across the solar air heater to evaluate the best performance parameters. The structure of this paper is as follows. First, we describe the approach to the theoretical modelling technique used to guide the development of the solar air heater, before describing the experimental equipment (the SAH) that was built and discussing the performance evaluation and further development of the SAH. This is the first time that the performance parameters i.e., heat transfer rate, air velocity, and baffle arrangements of SAH were optimized using the CFD approach.”
Q.2 Literature review is elaborated for each previous study, which is not required. Present only key results of the previous study in one or two lines only.
Ans.2 The literature review is shorter than the results only of the previous study as per directions.
Q.3 Also, I suggest including a more recent paper on solar air heaters relevant to your study, 8 papers are not sufficient in the introduction.
Ans.3 The suggestion is very valid so the number of relevant papers in the introduction is increased to 20.
Q.4 Include Grid independent Test of the study.
Ans.4 The grid-independent study is not possible because of the mass flow rate of 0.01 kg. s-1 is required for a spray drying system which can not be achieved by natural circulation. Forced circulation is required and it’s only possible from the blower. The spray drying system requires 150 C temperature and SAH is a medium-range temperature technology, so it is impossible to get 150 C from SAH. The results show that SAH decreased 30 % of the total electricity cost.
Q.5 The objective of the paper should be included at the end of the introduction.
Ans.5 The objective of the paper is added at the end of the introduction with the novelty of the design.
Q.6 Text in some figures is not visible, especially in Fig. 9. And 11.
Ans.6 New more visible figures 9 and figure 11 are added.
Q.7: Figure 10. Is missing.
Ans.7: There is a typing error now the numbering of each figure is corrected.
Q.8: Lines 376-377, The mass flow rate was 0.01 kg. s-1 and an increase in temperature cause the rise in internal energy and enthalpy as the temperature drops the enthalpy also decreased. This sentence has no meaning pls modified it.
Ans.8: The above-mentioned line is removed as it was not required here.

Reviewer 3 Report
In this paper , a kind of solar air heater is proposed to analyze its performance parameters and economic cost. This paper looks like a technique report about a solar air heater applied in agricultural products. There some questions here for authors as following:
1 The novelty of current research need more support information and be clearly described.
2 The authors designed a solar air heater for spray drying of milk powder, the unit structure and material were described in detail, but just show readers a geometry model for CFD, please give more real equipment information in the section of unit discussion.
3 The authors investigated the performance of this solar air heater by numerical and experimental method, but the target of numerical and experimental work isn’t present well, such as, what relationship between the performance obtained and previous mentioned parameters in Table 1 and Table 2 can be revealed by simulation or experiment. Please complement related discussion. And how to obtain the current operation conditions for current heater needs to be evaluated.
4 The similar question exist in section 3.2 and 3.3, the performance need related with previous parameters mentioned.
5 In line 315 , line 350, and line 390, as shown in Figure 7, Figure 9, and Figure 11, there are some curves about inlet temperature with time. Why the inlet temperature increased with time shown by these curves, as shown in table 3, the inlet temperature seems fixed in a experiment, please give more details.
6 This paper discussed economic analysis about the solar air heater, but it seems that the data for economic analysis is independent from previously discussed data of current solar air heater, please give more discussion.
Author Response
1 The novelty of current research needs more supporting information and to be clearly described.
Ans. 1 The Paragraph about the novelty of the research is added to the end of the introduction at line 155 to line 164.
“The aim of this work was to design and develop a drying SAH to couple with a milk powder spray drying system for converting raw milk to powder. We aimed to use computational fluid dynamics (CFD) to predict the airflow pattern and temperature distribution across the solar air heater to evaluate the best performance parameters. The structure of this paper is as follows. First, we describe the approach to the theoretical modelling technique used to guide the development of the solar air heater, before describing the experimental equipment (the SAH) that was built and discussing the performance evaluation and further development of the SAH.”
2 The authors designed a solar air heater for spray drying of milk powder, the unit structure and material were described in detail, but just show readers a geometry model for CFD, please give more real equipment information in the section of the unit discussion.
Ans. 3 Figure 9. An experimental setup for evaluating the performance parameters of SAH was added at line 422 page 12.
3 The authors investigated the performance of this solar air heater by the numerical and experimental methods, but the target of numerical and experimental work isn’t presented well, such as, what relationship between the performance obtained and previously mentioned parameters in Table 1 and Table 2 can be revealed by simulation or experiment. Please complement the related discussion. And how to obtain the current operating conditions for the current heater needs to be evaluated.
Ans.3 At the mass flow rate of 0.01 kg. s-1 and 150°C temperatures were required for the spray drying system. The experiment to convert raw milk into powder was carried out at 0.005 kg. s-1 and 0.01 kg. s-1. But the result with a mass flow rate of 0.005 kg. s-1 was not satisfactory. The CFD simulation of the SAH was done to check the airflow pattern and temperature distribution at the mass flow rate of 0.005 kg. s-1 and 0.01 kg. s-1 after baffles arrangements. The estimation of temperature at the SAH outlet was obtained after applying the boundary conditions (Solar irradiance, Heat loss coefficient of polystyrene, thermal conductivity, transmissivity of polycarbonate sheet, and absorptivity of absorber plate). The main target of the CFD simulation is to investigate the airflow pattern to minimize the blank spots and to check whether it is possible to use a locally fabricated SAH that can be used as a preheater. Table 1 and Table 2 parameters were used to estimate the parameters after the experimentation. The performance of the current solar heater can be obtained by the parameters presented in Table 1 and Table 2. Every SAH performance will be different because of the material used for construction (in terms of optical and thermal losses)
4 The similar question exists in sections 3.2 and 3.3, the performance needs related to the previous parameters mentioned.
The output figures mentioned in 3.2 and 3.3 have now been elaborated with help of the relevant equations used and details have been incorporated in the respective sections accordingly.
5 In lines 315, line 350, and line 390, as shown in Figure 7, Figure 9, and Figure 11, there are some curves about inlet temperature with time. Why does the inlet temperature increase with time shown by these curves, as shown in table 3, the inlet temperature seems fixed in an experiment, please give more details.
The Inlet temperature was on the y-axis with minimum coordinates difference. The Data taker data logger was used to measure every minute temperature difference. So the little fluctuation shows a huge change. However, the variation shows the sensitivity analysis of temperature profiles during the experiment. The temperature on the experiment table was taken as the average temperature of all the measured readings.
6 This paper discussed the economic analysis of the solar air heater, but it seems that the data for economic analysis is independent of previously discussed data of the current solar air heater, please give more discussion.
The fixed cost and variable cost were not mentioned in the previously discussed data which is now added on lines 255 and 275 respectively to address your query.

Round 2
Reviewer 1 Report
The authors have made a significant improvement to the paper. It is acceptible in its present form.
Author Response
- The authors have made a significant improvement to the paper. It is acceptable in its present form.
Ans. Many thanks for this greatly appreciated comment.

Reviewer 3 Report
1. The authors present a figure about experimental unit, but the detailed structure of this unit still can’t be described clearly to help understanding the importance of material in current SAH. Figure 9 didn’t appear in the text, And is Figure 8in Line 426 a writing error?
2. And another question is about the operations, air flow rate of 0.005 kg. s-1and 0.01 kg. s-1 are selected in CFD simulation and experiment. Just these tow flow rate are considered, are these enough to obtain current conclusion such as line 24 “An air mass flow rate of 0.01 kgs-1 is was required”.
3. The legend of Figure 3 and Figure 4 still need to improve to clearly present the difference in flow field.
4. It is still hard to understand correlation between the previous performance data analyzed in 3.2,3.3 section and the data analysis in 3.4 section in current version.
Author Response
- The authors present a figure about the experimental unit, but the detailed structure of this unit still can’t be described clearly to help understand the importance of the material in the current SAH. Figure 9 didn’t appear in the text, and in Figure 8in Line 426 a writing error?
Ans.1 The experimental unit is a solar-heated heat exchanger with baffles to create a serpentine pathway for the gas. The baffles increase the gas velocity and hence the heat-transfer coefficients from the absorber material to the gas, and the baffles also increase the heat-transfer area between the gas and the absorber surface. The materials were chosen for ease of construction in developing countries, such as Pakistan, so these materials are important due to their availability and low cost, as well as their reasonable heat-transfer performance mentioned in lines 188 to line 193. Figure 9 appears in line 397.
- And another question is about the operations, air flow rate of 0.005 kg. s-1and 0.01 s-1 are selected in CFD simulation and experiment. These two flow rates are considered, and are these enough to appear at ended current conclusion such as line 24 “An air mass flow rate of 0.01 kgs-1 was required”.
Ans 2. An air mass flow rate of over 0.005 kg/s is required for the reasonable performance of the spray dryer. Otherwise, the spray-drying performance is inadequate.
- The legend of Figure 3 and Figure 4 still need to improve to clearly present the difference in the flow field.
Ans 3. The legends of Figures 3 and 4 have been modified to clarify the difference, as mentioned below.
“Figure 3. Performance prediction of an air-flow distribution pattern using CFD for the SAH.”
“Figure 4. Performance prediction of air temperature distribution pattern using CFD for the SAH.”
- It is still hard to understand the correlation between the previous performance data analyzed in the 3.2,3.3 sections and the data analysis in the 3.4 section in the current version.
Ans 4. A paragraph to understand the connection between the previous performance data and the data analysis in section 3.4 has been added to line 190, as mentioned below.
“The structure of this section is to report and discuss the theoretical performance of the SAH in section 3.1, before discussing the actual heat losses and leakages from the experimental unit in section 3.2. The complete technical performance of the SAH and its further development follows in section 3.3, building on the important preliminary theoretical work in section 3.1 and practical foundations in section 3.2. To complete a techno-economic assessment of the SAH, the economics of the SAH are presented and discussed in section 3.4”.
